# Investigating foods and beverages sold and advertised in deprived urban neighbourhoods in Ghana and Kenya: a cross-sectional study

Mark Alan Green ,[1] Rebecca Pradeilles,[2] Amos Laar,[3] Hibbah Osei-Kwasi,[4] Nicolas Bricas,[5] Nathaniel Coleman,[6] Senam Klomegah,[7] Milka Njeri Wanjohi,[8] Akua Tandoh,[6] Robert Akparibo,[9] Richmond Nii Okai Aryeetey,[6] Paula Griffiths,[2] Elizabeth W Kimani-Murage,[8] Kobby Mensah,[10] Stella Muthuri,[11] Francis Zotor,[7] Michelle Holdsworth[12]

For numbered affiliations see end of article.

**Correspondence to**
Dr Mark Alan Green;
mark.green@liverpool.ac.uk

## ABSTRACT

**Objectives** The aim of this study was to characterise the local foods and beverages sold and advertised in three deprived urban African neighbourhoods.

**Design** Cross-sectional observational study. We undertook an audit of all food outlets (outlet type and food sold) and food advertisements. Descriptive statistics were used to summarise exposures. Latent class analysis was used to explore the interactions between food advertisements, food outlet types and food type availability.

**Setting** Three deprived neighbourhoods in African cities: Jamestown in Accra, Ho Dome in Ho (both Ghana) and Makadara in Nairobi (Kenya).

**Main outcome measure** Types of foods and beverages sold and/or advertised.

**Results** Jamestown (80.5%) and Makadara (70.9%) were dominated by informal vendors. There was a wide diversity of foods, with high availability of healthy (eg, staples, vegetables) and unhealthy foods (eg, processed/fried foods, sugar-sweetened beverages). Almost half of all advertisements were for sugar-sweetened beverages (48.3%), with higher exposure to alcohol adverts compared with other items as well (28.5%). We identified five latent classes which demonstrated the clustering of healthier foods in informal outlets, and unhealthy foods in formal outlets.

**Conclusion** Our study presents one of the most detailed geospatial exploration of the urban food environment in Africa. The high exposure of sugar-sweetened beverages and alcohol both available and advertised represent changing urban food environments. The concentration of unhealthy foods and beverages in formal outlets and advertisements of unhealthy products may offer important policy opportunities for regulation and action.

## Strengths and limitations of this study

► Detailed geospatial data of three deprived neighbourhoods of African cities.
► Inclusion of food and beverages sold and advertised by outlet which are often not collected.
► Multidimensional analyses consider the interactions between outlet types, food and beverages sold and advertised rather than treating factors independently.
► Limitations include descriptive analyses, lack of representativeness of cities, cross-sectional data and lack of investigation of which environmental features matter for diet.

and 6.3% and 15.2%, respectively among Kenyan adults.[1] This emerging issue worries policy-makers because of the growing prevalence of nutrition-related non-communicable diseases (NR-NCDs).[2–4] The generalised trends in dietary changes, termed the nutrition transition, include increased consumption of fat, particularly vegetable and edible oils; increased added sugar; increased animal-source foods and decreases in cereals and fibre, specifically in coarse grains, staple cereals and pulses.[5 6] The exact nature of changes in dietary behaviours, and the foods that drive the nutrition transition, vary according to region.[7]

Coinciding with these trends in obesity, many African cities have experienced rapid urban growth.[8 9] The rapid growth and upheaval of urban environments has resulted in major changes to the built environment including the provision of shops and services.[10–13] This has the potential to shape the local food environment that individuals are exposed to, defined here as the physical

## INTRODUCTION

Rising prevalence of overweight and obesity has been seen in most African countries. For instance, the prevalence of obesity in Ghanaian adults (≥20 years) in 2013 was 8.1% and 14% for men and women, respectively,

services, outlets and structures that promote, advertise and sell foods and beverages. It includes outlets that sell fresh, prepackaged and cooked foods and beverages such as local vendors, small shops, supermarkets, but also advertisements relating to foods and beverages. Local street vendors are key sources for purchasing foods and beverages to cook or eat at home, as well as to eat on site.[14] While there are few multinational organisations (eg, restaurant chains) established in African cities, they are increasingly becoming interested in the African market where eating out is a sign of affluence.[15] Economic growth has resulted in greater disposable income, which is increasingly being spent on foods and beverages. There has also been growth in supermarkets providing a greater variety of foods and beverages, although such stores are not commonly found within poorer neighbourhoods.[11 12 16 17]

Our understanding of the urban African food environment is limited. There are few studies from Africa (mainly concentrated in South Africa[18]), although studies are increasingly exploring urban food environments in middle-income countries which have some parallels.[19 20] This is despite extensive investigation of the nature, role and impact of food environments on diet and obesity in high-income countries.[21–24] The lack of ecological and environmental research has been identified as a key barrier towards developing successful interventions by policy makers.[25] This is important since obesity prevalence is higher in urban Africa (including in Ghana and Kenya).[15 26]

Most studies in high-income countries have focused on the location of specific types of outlets.[23] Few studies have measured the types of food and beverages sold by outlets that can allow generalisations about whether they are healthy or not.[19 27] Advertisements are rarely incorporated into analyses despite being ever-present features of most environments. Only focusing on outlet type alone and ignoring these broader characteristics limits our ability to build detailed measures of food environments to truly assess their influences on people.[21 27 28] We extend previous approaches by using a multidimensional approach to measure the location of food outlets and adverts, as well as what food and beverages are being sold or advertised. Exploring the interactions between these factors is important to characterise and therefore monitor the food environments populations are exposed to.[29]

The aim of our study was to characterise the local foods and beverages sold and advertised in three deprived urban African neighbourhoods.

## METHODOLOGY
### Data
Three residential neighbourhoods were selected: Jamestown in Accra (Accra population ~1.5 million), Ho Dome in Ho (Ho population ~100 000) both in Ghana, and Makadara in Nairobi, Kenya (Nairobi population ~3.5 million). Online supplementary appendix A presents details and maps of the three study sites. They were each identified by randomly selecting one deprived neighbourhood or community in each city that was not a slum based on the following sources: in Accra, we were informed by the findings of the Accra Poverty Mapping exercise[30]; in Ho, we used data from the United Nations Human Settlements Programme[31]; finally, in Nairobi we used ward-level deprivation estimates from the Kenya National Bureau of Statistics.

We conducted a full audit of each neighbourhood. Community audit approaches are common in studies of the food environment and allow the construction of detailed and focused datasets.[21] Neighbourhoods were defined through discussion with local community groups, and a 250 m buffer was placed around the edge of the community to minimise edge effect issues.

The data collection method was piloted in each setting through surveying smaller areas close to where data collection would take place. The tool was tested on each outlet types (number of occasions for each depending on prevalence in an area) and further amendments were made to the tool. Data collection occurred between September and December 2017. Trained researchers recorded any food outlet or advert observed within each study site. We recorded the outlet type using a classification we developed during a project partners workshop (a full list of outlets including descriptions can be found in online supplementary appendix B). We defined outlet types as 'informal' if the structure of their shop (if any) was movable, not permanent and/or a small-scale operation. Items sold within outlets were recorded based on a predefined framework of how they would be expected to change during the nutrition transition (detailed in online supplementary appendix B). All food advertisements within an outlet were also recorded. We recorded the type of advert (see online supplementary appendix B for description) and the item it was advertising (using same categories as the items sold). We also recorded any standalone adverts including the type of advert and the food advertised. Global Positioning System coordinates of outlet and advert locations were recorded using Garmin handheld devices.

### Statistical analysis
Descriptive statistics were used to explore the characteristics of our samples. Latent class analysis was used to identify groups within our data to characterise the food environment. Latent class analysis is useful for identifying subgroups within categorical data that are otherwise unknown.[32] An exploratory approach was used in the absence of an a priori understanding of what groups to expect. We included the outlet type, items being sold, advert type and items being advertised in the analytical models; although this represents a large number of variables, we had no prior justification for dropping variables in this exploratory analysis. The number of latent classes were evaluated through assessing model fit. Analyses were completed using R statistical software (V.3.6.2).[33]

**Table 1** Outlet type and items sold by neighbourhood

| Groups | Measure | Jamestown | Ho Dome | Makadara | All |
|---|---|---|---|---|---|
| | *Outlet type (percentage of all outlets)* | | | | |
| Formal | Bar/pub | 8.8 | 5.0 | 6.6 | 7.1 |
| | Restaurant | 0.0 | 2.8 | 3.3 | 2.0 |
| | Supermarket | 0.5 | 0.6 | 1.3 | 0.9 |
| | Shop | 10.1 | 43.9 | 17.9 | 19.6 |
| | All formal outlets | 19.4 | 52.3 | 29.1 | 29.6 |
| Informal | Kiosk | 15.1 | 3.9 | 40.0 | 24.3 |
| | Local vendor | 1.6 | 22.8 | 13.4 | 10.6 |
| | Vegetable/fruit stand/table top | 63.9 | 21.1 | 17.5 | 35.6 |
| | All informal* outlets | 80.5 | 47.8 | 70.9 | 70.5 |
| | *Foods and beverages sold (percentage of all outlets)* | | | | |
| Expected to increase during the nutrition transition | Fats/oils | 13.5 | 43.3 | 28.7 | 25.5 |
| | Sugar-sweetened spreads | 6.5 | 49.4 | 25.6 | 22.6 |
| | Fresh meat/poultry | 16.9 | 22.8 | 14.4 | 16.8 |
| | Fresh fish/shellfish | 14.6 | 17.8 | 7.2 | 11.8 |
| | Milk | 24.9 | 43.3 | 29.3 | 30.1 |
| | Eggs | 27.8 | 53.9 | 37.0 | 36.5 |
| | Sugar-sweetened beverages | 35.8 | 50.0 | 37.4 | 39.0 |
| | Alcohol | 16.4 | 26.1 | 8.3 | 14.5 |
| | Processed/fried foods | 37.4 | 68.3 | 36.3 | 42.4 |
| | Cakes/sweets | 23.1 | 50.0 | 32.2 | 31.9 |
| | Modern mixed dishes | 8.1 | 4.4 | 1.8 | 4.6 |
| | Condiments | 15.1 | 43.3 | 22.5 | 23.4 |
| Expected to decrease | Staples | 25.7 | 55.9 | 37.0 | 39.8 |
| | Legumes/pulses | 8.6 | 25.6 | 21.9 | 17.5 |
| | Nuts/seeds | 9.1 | 38.3 | 15.5 | 17.1 |
| | Traditional dishes | 29.9 | 30.0 | 16.2 | 23.8 |
| | Fruits | 2.9 | 6.1 | 24.1 | 12.9 |
| | Vegetables | 19.5 | 21.7 | 37.9 | 28.1 |

*Informal/formal outlets were defined by authors and relate to whether the establishment is a long-term structure that could be regulated or not, or was small-scale in investment/organisation.

## RESULTS

### Describing the food environments

A total of 413 food outlets or adverts were identified in Jamestown, 208 in Ho Dome and 499 in Nairobi. We removed 53 observations that were outlets coded as other, resulting in an analytical sample of 1067. Descriptive statistics of the sample are presented in tables 1 and 2.

Each of the neighbourhoods contained different concentrations of outlet types (table 1). While the majority of outlets in both Jamestown (80.5%) and Makadara (70.9%) were defined as informal vendors, the nature of outlet types varied by neighbourhood. Jamestown was dominated by vegetable/fruit stand/table tops (63.9%), with low-availability of all the other types of outlets apart from kiosks (15.1%). Makadara was more evenly spread between kiosks (40.0%), local vendors (13.43%),

vegetable/fruit/food stand/table top (17.5%) and shops (17.9%). Ho Dome was different with the majority of outlets defined as formal vendors. There was high availability of shops (43.9%), as well as local vendors (22.8%) and vegetable/fruit/food stand/table top (21.1%).

Foods and beverages sold in outlets demonstrated a good diversity of items. Outlets on average offered a mixture of healthy and unhealthy items (see online supplementary appendix C for more details). There was high availability of sugar-sweetened beverages (39%), as well as processed/fried foods (42.4%). Most items were more common in Ho Dome. This reflected the differences in the prevalence of outlets types, with a higher prevalence of formal vendors that sold a greater number of items. Informal outlets had greater availability of healthier foods and fewer alcohol adverts compared with

**Table 2** Food and beverage advertisements observed by neighbourhood, type and items advertised

| | Jamestown | Ho Dome | Makadara | All |
|---|---|---|---|---|
| *Whether outlets contained an advert (percentage of all records)* | | | | |
| Outlet with advert(s) | 25.3 | 39.1 | 22.7 | 26.6 |
| Outlets with no advert | 70.2 | 58.7 | 76.9 | 71.1 |
| Standalone advert | 4.5 | 2.2 | 0.4 | 2.3 |
| *Advert type (percentage of outlets with adverts or standalone adverts)* | | | | |
| Billboard | 0.0 | 7.9 | 2.8 | 3.0 |
| Poster | 78.3 | 57.9 | 50.9 | 63.6 |
| On site | 24.2 | 36.8 | 50.0 | 36.4 |
| Painting | 19.2 | 23.7 | 10.4 | 17.2 |
| *Foods and beverages advertised (percentage of outlets with adverts or standalone adverts)* | | | | |
| Fats/oils | 0.8 | 5.3 | 7.6 | 4.3 |
| Sugar-sweetened spreads | 2.5 | 1.3 | 6.6 | 3.6 |
| Fresh meat/poultry | 5.8 | 15.8 | 8.5 | 9.3 |
| Fresh fish/shellfish | 2.5 | 11.8 | 1.9 | 4.6 |
| Milk | 11.7 | 30.3 | 30.2 | 22.9 |
| Eggs | 3.3 | 0.0 | 7.6 | 4.0 |
| Sugar-sweetened beverages | 57.5 | 34.2 | 48.1 | 48.3 |
| Alcohol | 34.2 | 31.6 | 19.8 | 28.5 |
| Processed/fried foods | 6.7 | 9.2 | 11.3 | 8.9 |
| Cakes/sweets | 1.7 | 4.0 | 18.9 | 8.3 |
| Modern mixed dishes | 4.2 | 6.6 | 1.9 | 4.0 |
| Condiments | 8.3 | 15.8 | 6.6 | 9.6 |
| Staples | 5.8 | 23.7 | 16.0 | 13.9 |
| Legumes/pulses | 0.8 | 1.3 | 2.8 | 1.7 |
| Nuts/seeds | 0.0 | 0.0 | 4.7 | 1.7 |
| Traditional dishes | 5.8 | 11.8 | 3.8 | 6.6 |
| Fruits | 0.0 | 1.3 | 2.8 | 1.3 |
| Vegetables | 4.2 | 2.6 | 4.7 | 4.0 |

formal vendors. There was lower prevalence of alcohol sold in Makadara. Fresh meat, poultry and fish were all more common in Jamestown and Ho Dome, with fruits and vegetables more common in Makadara.

A quarter of outlets contained advertisements and with the highest percentage in Ho Dome (table 2; see online supplementary appendix C for more details). Few standalone adverts were observed across each location. The low proportion of billboards reflects that data collection was conducted in residential neighbourhoods, with fieldwork revealing that billboards were mainly found alongside main roads. Paintings were more commonly observed in Jamestown and Ho Dome than Makadara. Posters were the most common type of advert found (63.6%). 'On site/front of outlet' advertisements were also fairly common especially in Makadara, with large scale paintings less prevalent.

On average, the types of foods and beverages advertised were mainly different from those sold by outlets. The most common item advertised was sugar-sweetened beverages (48.3%) and this was consistent across each neighbourhood (including over half of all occasions where adverts were observed in Jamestown). Milk products were also more commonly advertised than most other items. Alcohol was also widely advertised in Jamestown and Ho Dome, but not in Makadara.

### Typology of the food environment

We sought to examine the interactions in our measures and classify the food environment. A five-class model was selected as the most appropriate typology (see online supplementary appendix D). To help our interpretation of each model, figure 1 presents radial plots of the conditional response probabilities that describe class characteristics. Where lines reach outwards, they represent higher probabilities and therefore a class being more likely to contain that particular variable. We have provided qualitative names for each class to further help their interpretation.

Interpretation of the latent classes:

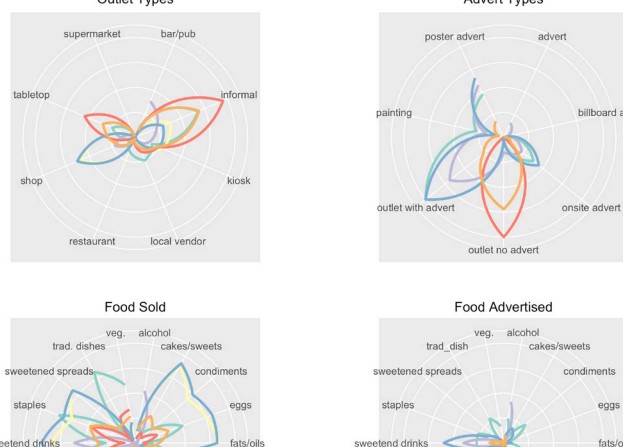

**Figure 1** Conditional response probabilities of the latent class model.

1. *Informal outlets selling raw ingredients*: it was the smallest class ($\gamma$=0.06). There was an even spread of outlet types observed, although the majority were informal vendors ($\rho$=0.68). All outlets contained adverts, with higher proportions of posters ($\rho$=0.48), on site ($\rho$=0.38) and paintings ($\rho$=0.28). Foods sold were higher for traditional raw and unprocessed ingredients including staples ($\rho$=0.81), traditional dishes ($\rho$=0.82), fresh meat/poultry ($\rho$=0.69) and eggs ($\rho$=0.50) (although note the high rate of processed/fried foods as well; $\rho$=0.72). Advertisements largely followed the foods and beverages sold.

2. *Formal outlets selling unhealthy foods and beverages with no adverts*: the class was fairly small ($\gamma$=0.13). The majority of outlets were shops ($\rho$=0.60), with the remaining proportion mostly made up of kiosks ($\rho$=0.30). There was high availability of unhealthy foods sold particularly items typically added to the cooking process such as fats/oils ($\rho$=0.83), sugar-sweetened spreads ($\rho$=0.83) and condiments ($\rho$=0.68). Sugar-sweetened beverages ($\rho$=0.84) and cakes/sweets ($\rho$=0.89) were also common. There was good availability of some healthier foods such as milk ($\rho$=0.85), eggs ($\rho$=0.76) and staples ($\rho$=0.67). There was low-availability of fresh meat, poultry, fish, fruit or vegetables. There were no advertisements.

3. *Drinking establishments*: a relatively small class ($\gamma$=0.14). Outlets were mainly formal vendors and the majority contained adverts. The class had the highest prevalence of pubs/bars ($\rho$=0.39), as well as standalone adverts ($\rho$=0.16). Adverts were mainly posters ($\rho$=0.58), although there were above average proportions for the other advert types. The class was characterised by

the highest proportion of alcohol and sugar-sweetened beverage sold and advertised. There were few other items sold or advertised.

4. *Small informal sellers*: the largest class ($\gamma$=0.57) was mostly informal outlets ($\rho$=0.95) with no adverts. It had the highest prevalence of vegetable/fruit stand/table top ($\rho$=0.55). There was low-availability of most items. This reflects sellers having a low diversity of items, selling only a few items.

5. *Formal outlets selling unhealthy foods and beverages with adverts*: the final class was also fairly small ($\gamma$=0.11). The characteristics of the class largely followed class 2, with the noticeable difference being that all outlets contained adverts. Foods and beverages that were advertised matched what was being sold. It had the highest proportion of posters ($\rho$=0.66) and on-site advertisements ($\rho$=0.45).

We stratified our analyses by neighbourhood; however, the results were similar suggesting that our classes were consistent by context (results not reported).

## DISCUSSION

Our study presents one of the most detailed investigations of accessibility and exposure to food and beverages sold and advertised within deprived neighbourhoods of African cities. Our findings have relevance both for understanding the food environments within the Ghanaian and Kenyan context, and could be transferable to other African contexts and beyond. We find diverse food environments across each neighbourhood, but also broad similarities. Other than in Ho Dome, outlets were mainly informal vendors (eg, kiosks or vegetable/fruit stands/table tops). While there was high availability of many healthy foods, there was considerable availability of unhealthy, so called energy-dense nutrient-poor foods and beverages (eg, sugar-sweetened beverages, fats/oils, processed/fried foods). Advertisements were most frequent for sodas or sweetened beverages, alcohol and milk, with low-advertising for other items (particularly healthy foods and beverages). These findings require attention since the availability and promotion of unhealthy energy-dense nutrient poor foods and beverages within deprived urban African environments may place considerable burdens on health systems. Poor dietary intake and alcohol abuse are key drivers of NCDs accounting for 73.4% of all deaths globally.[4] With ~80% of NCDs occurring in low-income and middle-income countries and an increasing burden in sub-Saharan Africa,[3] the consequences of unhealthy environments are potentially considerable.

Our data reveal good physical availability of food and beverages across three deprived neighbourhoods. We do not find evidence of food deserts in contrast to social inequalities in high-income countries.[21 24 34] Availability is made possible through the informal sector; local sellers are often mobile allowing them to take opportunities of free space to sell items. The growth of informal outlets reflects the rapid urbanisation and change experienced

in cities,[8 9 13] and with respect to the food environment our findings suggest that the growth of informal outlets may not necessarily be bad. The greater availability in unhealthy foods and beverage sold and advertised within formal outlets may offer potential strategies to policy makers. Given that they have physical structures (unlike informal outlets), formal outlets may be amenable to planning regulations (eg, restricting location, rent subsidies) and interventions to promote healthier foods (eg, price subsidies, promoting healthier products).[35] The same strategy will apply also to advertisements particularly given their focus on sugar-sweetened beverages and alcohol.[36] Such recommendations contrast with experiences in many African countries whereby the state often regard formal actors as having positive influences on food systems and therefore are less interested in any form of regulation.[11 34]

While we find good physical availability of healthy foods, this represents only one aspect of accessibility. There was good availability of unhealthy foods and beverages that are energy-dense and nutrient poor nutrient as well. Exploring the relationship between availability and consumption of foods and beverages represents an important future angle for research.[20 29 37] Acceptability and affordability also represent important domains of accessibility. Our data would suggest that focusing on improving accessibility to healthy foods and beverages only through physical availability may not be effective since such items are commonly available. Improving the acceptability or desirability of healthy foods and beverages can only go so far as well. Affordability is paramount; for example, achieving five servings of fruit and vegetables per person per day would cost 52% of household income in low income countries.[38] Making healthy food affordable, as well as available, will be a key area for policy.[12 34]

The higher exposure to sodas or sugar-sweetened beverages in the items sold and advertised was consistent across each location. Such products are often associated with social status, particularly if the product originates from high-income countries.[39] They also reflect considerable efforts by large multinational corporations to increase sales of sugar-sweetened beverages in African nations through investing in infrastructure including adverts.[14] Investment has been pronounced and its effectiveness, when combined alongside television and sponsorship campaigns to improve the acceptability of their products, has altered the food environments of urban areas.[40] Countering these messages through increased education and making healthier alternatives more desirable could be effective; however, public health budgets are dwarfed by the marketing budgets of such companies suggesting it may be difficult. Given the association between sugar consumption and NR-NCDs,[5 41] this area represents an urgent policy area.

Alcohol also displayed higher exposure through items sold and advertised. Africa is an emerging market for alcohol producers and companies are using advertisement campaigns to increase the exposure to alcohol impacting on the built environment.[36 42] Tackling such advertisements represents one potential important area for policy. There was less alcohol available and advertised in Kenya than compared with Ghana. This was despite a similar proportion of bars/pubs in each context. While alcohol consumption is far lower in Africa than Europe, for example, recent estimates suggest it Is slightly higher in Kenya compared with Ghana.[42] Alcohol licensing laws are stricter in Kenya, with establishments recently required to have a formal liquor sale licence to sell any alcohol (with no sales during the day). It suggests that regulation can be effective in minimising exposure to alcohol, and might lead to opportunities to tackling other unhealthy food items.

A strength of our study is our multi-dimensional approach for measuring the interactions across the food environment. Previous studies tend to focus on measuring single aspects of the food environment and their relationship to health outcomes independently. While focusing on specific factors is useful for isolating dietary components that could be targeted with interventions,[22] they ignore how features interact together and therefore are reductionist in how they measure the food environment.[21 27 28] For example, the latent classes reveal how the foods sold and advertised differ considerably by outlet type. Our approach offers one way for the effective monitoring of food environments, which is important for targeting interventions.[29] The unstructured urban growth that most African cities are experiencing has made it difficult for governments to regulate environments. Without effective monitoring and targeting systems in place, the continued unregulated growth of urban areas may act as a catalyst for rising prevalence of obesity and NR-NCDs.[10]

There are several limitations to our study. Although our approach provides a nuanced way to measure the food environment, it is not all encompassing. While we included a lot of detail in our measures, important aspects of food supply were not collected (eg, mobile food vendors). We did not consider the role of urban farming of crops and animals.[43] Product placement, shelf space, price and promotional offers are each stimuli influencing food choices within outlets.[27] Data were static representing a single time point, but African food environments evolve through the day, and are vibrant and busy at night. Food hygiene and sanitation surrounding outlets is an important determinant of food borne diseases and diarrhoea. We also focused on the local neighbourhood environments, but this ignores the wider environments that individuals engage in during their days, for example, school environments.[37] Extending our approach towards developing multisite and multiscale measures will help to build more nuanced measures that incorporate a greater range of contexts that may influence dietary behaviours. Our study though provides a useful starting point for developing more nuanced and detailed understanding of the food environment.

A further limitation is that while our study sites provide insightful findings of the local food environment, each

location is not representative of its city. The rapid growth of many African cities has been unplanned, unstructured and almost random resulting in diverse and heterogeneous communities.[44] Extending study sites to a greater range of neighbourhoods will help to evaluate the interpretation our findings. Our data are cross-sectional and could have been improved through longitudinal observations to explore the consistency of our findings. This is a common issue in studies of the food environment.[22–24] Finally, our study is purely descriptive of food environments and future studies should extend our analyses to explore the extent that exposure to food sold and advertised influence dietary behaviours.

## CONCLUSIONS

Our study presents one of the most detailed explorations of the food environment in African cities. The physical built environment is a largely ignored aspect of food policy focus in the urban African context; however, our findings demonstrate the need to consider the geographical context relating to the foods and beverages sold and advertised within neighbourhoods. Healthy foods commonly available within each neighbourhood suggests that focusing on availability alone may not be effective. Regulating formal marketing and advertisements may be more appropriate, and understanding how to effectively intervene to reduce their exposure will be important for effective policy implementation.

**Author affiliations**
[1]Department of Geography & Planning, University of Liverpool, Liverpool, UK
[2]School of Sport, Exercise and Health Sciences, Loughborough University, Loughborough, UK
[3]Department of Population, Family & Reproductive Health, School of Public Health, University of Ghana, Accra, Ghana
[4]Department of Geography, The University of Sheffield, Sheffield, UK
[5]UMR MOISA, CIRAD, Montpellier, France
[6]Department of Population, Family and Reproductive Health, University of Ghana, Legon, Ghana
[7]Department of Family and Community Health, University of Health and Allied Sciences, Hohoe, Ghana
[8]Maternal and Child Wellbeing Unit, African Population and Health Research Center, Nairobi, Kenya
[9]School of Health and Related Research (ScHARR), The University of Sheffield, Sheffield, UK
[10]Business School, University of Ghana, Legon, Ghana
[11]DFID Kenya, Nairobi, Kenya
[12]NUTRIPASS Unit, French Research Institute for Sustainable Development (IRD), Montpellier, France

**Contributors** MAG, AL, RA, PG, EWK-M, KM, FZ and MH had the idea for the study. All authors were involved in designing the data collection approach and tools. NC, SK, MNW and AT collected the data under the supervision of AL, EWK-M, SM and FZ. MAG, RP, HO-K, NB and MH undertook the statistical analyses, with all authors scrutinising the results. MAG led the writing of the paper, with all authors contributing to revising and approving the paper. The corresponding author (MAG) had full access to all the data in the study and had final responsibility for the decision to submit for publication.

**Funding** This work was supported by two funders. The 'Dietary Transitions in Ghana' project was funded by a grant from the Drivers of Food Choice (DFC) Competitive Grants Programme [grant number OPP1110043] which is funded by the Bill and Melinda Gates Foundation and the Department for International Development (DFID), and managed by the University of South Carolina Arnold School of Public Health, USA. DFC supports new research on understanding food choice among the poor in low/middle-income countries, strengthening country-level leadership in nutrition and fostering a global community of food-choice researchers. The TACLED project was funded by a Global Challenges Research Fund (GCRF) Foundation Award led by the MRC [grant number MR/P025153/1], and supported by AHRC, BBSRC, ESRC and NERC, with the aim of improving the health and prosperity of low/middle-income countries.

**Competing interests** None declared.

**Patient and public involvement** Patients and/or the public were not involved in the design, or conduct, or reporting, or dissemination plans of this research.

**Patient consent for publication** Not required.

**Ethics approval** Ethical approval for the study was acquired by each institution involved in the data collection process. In Ghana, ethics approval was obtained from the Ghana Health Service Ethics Review Committee (references: GHS-ERC 07/09/16 and GHS-ERC 02/05/17). In Kenya, ethics approval was obtained from the African Medical and Research Foundation (AMREF) (reference: ESRC P365/2017). The University of Sheffield recognised both of these approvals as meeting their ethical standards, as did Loughborough University for the Ghana Health Service. Additional ethical approval was obtained from the University of Liverpool (references: 1434 and 2288) and Loughborough University (reference: R17-P142).

**Provenance and peer review** Not commissioned; externally peer reviewed.

**Data availability statement** All data are openly available. Data for Jamestown and Ho Dome (Accra and Ho respectively, Ghana) can be accessed via https://doi.org/10.23708/QYHL8Chttps://doi.org/10.23708/QYHL8C. Data for Makadara (Nairobi, Kenya) can be accessed via https://doi.org/10.23708/G76QVS. All analytical code can be viewed at https://github.com/markagreen/Food_environment_ghana_kenya.

**ORCID iD**
Mark Alan Green http://orcid.org/0000-0002-0942-6628

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
