## [Reviewer comments · BMJ Open]

ARTICLE DETAILS

TITLE (PROVISIONAL)	Investigating foods and beverages sold and advertised in deprived urban neighbourhoods in Ghana and Kenya: A cross-sectional study
AUTHORS	Green, Mark; Pradeilles, Rebecca; Laar, Amos; Osei-Kwasi, Hibbah; Bricas, Nicolas; Coleman, Nathaniel; Klomegah, Senam; Wanjohi, Milka; Tandoh, Akua; Akparibo, Robert; Aryeetey, Richmond; Griffiths, Paula; Kimani-Murage, Elizabeth; Mensah, Kobby; Muthuri, Stella; Zotor, Francis; Holdsworth, Michelle

VERSION 1 – REVIEW

REVIEWER	Larissa Loures Mendes Universidade Federal de Minas Gerais Brazil
REVIEW RETURNED	02-Dec-2019

GENERAL COMMENTS	Minor Suggestions Perhaps it is more interesting for the typology of outlets to classify foods sold according NOVA system. The system consider to the degree of processing of foods and not just foods healthy and unhealthy. For example: "formal outlets selling unhealthy foods with no adverts" would be called "formal outlets selling predominantly ultra-processed foods"; "informal outlets selling raw ingredients" would be called "informal outlets selling unprocessed/minimally processed foods and processed culinary ingredients". The NOVA system is recognized by the World Health Organization, Food and Agriculture Organization, and the Pan American Health Organization. - I suggest that alcohol and sugary drinks should not be considered as food but as drinks. Thus it would be interesting to change the name of the group Food and drinks we expect to increase during the nutrition transition. Question - One question was regarding the condiments, are they natural or industrialized?
--

REVIEWER	Fernanda H Marrocos Leite Center for Epidemiological Research in Nutrition and Health, School of Public Health, University of São Paulo
REVIEW RETURNED	29-Dec-2019

GENERAL COMMENTS	General comments: The thematic of this manuscript is very relevant, considering that: 1) Diet-related diseases are on the rise in low- and middle-income countries, and are clearly linked with urban residence; 2) Food choices are affected not only by affordability and availability of
--

foods but also by other aspects of the food environment, such as desirability and convenience (significantly influenced by marketing); 3) there is limited knowledge about how people interact with food environments in LMICs to make food choices that shape nutrition and the risk for nutrition related NCDs. However, despite the manuscript relevance, I would recommend the authors to consider the following suggestions:

Specific comments:

TITLE: Please could you clarify the use of "food and advertising environments" instead of "food environments"? According to the ANH-FEWG: "The food environment is the interface that mediates one's food acquisition and consumption within the wider food system. It encompasses multiple dimensions such as the availability, accessibility, affordability, desirability, convenience, marketing, and properties of food sources and products". In my view, advertising is an element of food environments. I would recommend the authors to review this. See reference: Turner, C., Kadiyala, S., Aggarwal, A., Coates, J., Drewnowski, A., Hawkes, C., Herforth, A., Kalamatianou, S., Walls, H. (2017). Concepts and methods for food environment research in low and middle income countries. Agriculture, Nutrition and Health Academy Food Environments Working Group (ANH-FEWG). Innovative Methods and Metrics for Agriculture and Nutrition Actions (IMMANA) programme. London, UK.

ABSTRACT:

- Line 8, p. 2: I suggest replacing the text with "food environment of three deprived neighbourhoods in African cities"

INTRODUCTION

- Line 57-59, p. 4: I would recommend to review the following statement "or indeed in low- and middle-income countries in general". Although most part of the evidence is from high-income countries, studies are increasingly exploring urban food environments in middle-income settings. See a few examples provided in the comment below.

- Line 20-22, p. 5: I suggest referring to studies carried out in other lower-income countries which already measured the types of foods sold by outlets. See below a few examples from Brazil:

1) Leite et al. 2018. Association of neighbourhood food availability with the consumption of processed and ultra-processed food products by children in a city of Brazil: a multilevel analysis. *Public Health Nutr.* 2018 Jan;21(1):189-200.

2) Duran et al. 2013. Neighborhood socioeconomic characteristics and differences in the availability of healthy food stores and restaurants in Sao Paulo, Brazil. *Health Place.* 2013 Sep;23:39-47.

3) Leite et al. 2012. Availability of processed foods in the perimeter of public schools in urban areas. Availability of processed foods in the perimeter of public schools in urban areas. *J Pediatr (Rio J).* 2012;88(4):328-34.

- Lines 38-40, p. 5: I would recommend including the main hypotheses of the study/research question at the end of this section.

METHODOLOGY

- Line 17-20, p. 6: Please could you provide more details on the selection of neighbourhoods with local community groups? Which

	kind of groups? What were the particular characteristics of these groups in each of the selected cities?  - Lines 22-23, p. 6: Please give more information on the pilots (e.g. how many outlets were visited, was the tool validated for each setting?) - Lines 29-30, p. 6: Why have you decided to use a classification discussed during a project workshop instead of other tools? How were the categories chosen and who was involved in this workshop? - Line 36, p. 6: Which parameters/references were used to group foods expected to increase or decrease in consumption following the nutrition transition? Please clarify and give more detail in the methodology section. - Lines 29-30, p. 7: Please include the name and reference of the Statistical Software used in the analysis. - Line 20, p. 8: Include the total of all formal outlets - Lines 55-57, p. 8: Although the authors say: "outlets on average offered a mixture of healthy and unhealthy foods", Table 1 does not show which outlet types were more responsible or not for healthy or unhealthy food availability. It would be interesting to see the frequency of food types per outlet types (e.g. a table presented in the appendixes). - Lines 15-55, p. 9: Again, it would be interesting to see the types of foods advertised per outlet types (instead of per cities) as well (e.g. as an appendix). DISCUSSION  - Lines 15-19, p. 13: I would suggest the authors to expand on the consequences that this result could pose to the health system. 1) poor dietary intake and alcohol abuse are among key NCD risk factors, 2) NCDs are the leading cause of death globally, killing more people each year than all other causes combined, 3) 80% of NCDs occur in low- and middle-income countries. - Lines 40-42, p. 13: The authors should be present a few examples and compare with the literature. Although most of interventions to improve availability of healthier options in the community food environment are from high-income settings, there is still an opportunity to cite a few examples from the existing literature. E.g. Gittelsohn, et al. The Impact of a Multi-Level Multi-Component Childhood Obesity Prevention Intervention on Healthy Food Availability, Sales, and Purchasing in a Low-Income Urban Area. Int J Environ Res Public Health. 2017 Nov; 14(11): 1371. - Lines 33-38, p. 14: The authors could cite a few examples of increased investments from the SSBs multinationals in the African region (e.g. Taylor et al. 2016. Carbonating the World: The Marketing and Health Impact of Sugar Drinks in Low- and Middle-income Countries. Available at: https://cspinet.org/sites/default/files/attachment/Final%20Carbonating%20the%20World.pdf) - Line 44, p. 14: There are more recent references that could be used here (this one from 2011 is out of date). I recommend the authors to review a few citations and try to use more recent ones. CONCLUSIONS  - Lines 46-51, p. 16: "The physical built environment is largely ignored aspect...". Do the authors refer to the African context? There is extensive literature and food policy interventions looking at this aspect in high- and middle-income settings (See the NOURISH
--	---

VERSION 1 – AUTHOR RESPONSE

Reviewer: 1

Reviewer Name: Larissa Loures Mendes

Institution and Country:

Universidade Federal de Minas Gerais

Brazil

Please state any competing interests or state 'None declared': None declared

- Thank you for your comments and helpful suggestions.

Please leave your comments for the authors below

Minor Suggestions

Perhaps it is more interesting for the typology of outlets to classify foods sold according NOVA system. The system consider to the degree of processing of foods and not just foods healthy and unhealthy. For example: "formal outlets selling unhealthy foods with no adverts" would be called "formal outlets selling predominantly ultra-processed foods"; "informal outlets selling raw ingredients "would be called "informal outlets selling unprocessed/minimally processed foods and processed culinary ingredients". The NOVA system is recognized by the World Health Organization, Food and Agriculture Organization, and the Pan American Health Organization.

- We considered the NOVA classification system during the development of our methods and agree that is relevant to many food contexts where the food landscape is dominated by processed foods, such as in many high and upper-middle income countries. We decided that it was not relevant in the Ghanaian/Kenyan context. The classification of foods into "healthy" and "unhealthy" was informed by our related work on dietary intake (as part of the broader project; see Holdsworth et al., In Preparation), which used a nutrient profiling classification based on recognised methods (Drewnowski 2005) of foods consumed in the same cities. We found that some traditional foods and drinks were also energy dense and poor in nutrients regardless of processing. Therefore, as our related study found that consumption of packaged and commercially processed foods was low in all cities, the NOVA system was less relevant in the African context.
- We elected to focus on the nutritional transition since it is outlined from the outset within our theoretical framework and justification. To help strengthen this, we have added some addition text to the introduction: "*The generalised trends in dietary changes, termed the nutrition transition, include increased consumption of fat, particularly vegetable and edible oils; increased added sugar; increased animal-source foods and decreases in cereals and fibre, specifically in coarse grains, staple cereals and pulses.(5,6) The exact nature of changes in dietary behaviours, and the foods that drive the nutrition transition, vary according to region.*" (p4). It ensures our paper remains consistent in its message throughout. Therefore, we were interested to see whether foods that commonly increase/decrease in the context of nutrition transition were widely available and advertised. We have edited the text in the methodology to make this clear and guide the reader to see the greater detail in the appendix: "*Items sold within outlets were recorded based on a pre-defined framework of how they would be expected to change during the nutrition transition (detailed in Appendix B).*" (p6)
- We have clarified our food descriptions based on whether we classified them as 'healthy' or not in Table B3 (p5), including a short description of the decision-making process. This ensures the discussion of our results remain consistent, as well as helping to guide the reader throughout. We did not classify foods during data collection, however applied the classification after to aid the interpretation of our results.

- I suggest that alcohol and sugary drinks should not be considered as food but as drinks. Thus it

would be interesting to change the name of the group Food and drinks we expect to increase during the nutrition transition.

- We have added the term beverages to our title and aim, and we now refer to 'foods and beverages' throughout the text (and each Table) as well where relevant.

Question

- One question was regarding the condiments, are they natural or industrialized?

- They can include products that are commercially processed from multinational companies, as well as those prepared at home or with local small-scale production. We have included this statement in the appendix (p5).

Reviewer: 2

Reviewer Name: Fernanda H Marrocos Leite

Institution and Country: Center for Epidemiological Research in Nutrition and Health, School of Public Health, University of São Paulo

Please state any competing interests or state 'None declared': None declared

Please leave your comments for the authors below

General comments:

The thematic of this manuscript is very relevant, considering that: 1) Diet-related diseases are on the rise in low- and middle-income countries, and are clearly linked with urban residence; 2) Food choices are affected not only by affordability and availability of foods but also by other aspects of the food environment, such as desirability and convenience (significantly influenced by marketing); 3) there is limited knowledge about how people interact with food environments in LMICs to make food choices that shape nutrition and the risk for nutrition related NCDs. However, despite the manuscript relevance, I would recommend the authors to consider the following suggestions:

- Thank you for your positive comments and detailed suggestions on how to improve the paper.

Specific comments:

TITLE: Please could you clarify the use of "food and advertising environments" instead of "food environments"? According to the ANH-FEWG: "The food environment is the interface that mediates one's food acquisition and consumption within the wider food system. It encompasses multiple dimensions such as the availability, accessibility, affordability, desirability, convenience, marketing, and properties of food sources and products". In my view, advertising is an element of food environments. I would recommend the authors to review this. See reference: Turner, C., Kadiyala, S., Aggarwal, A., Coates, J., Drewnowski, A., Hawkes, C., Herforth, A., Kalamatianou, S., Walls, H. (2017). Concepts and methods for food environment research in low and middle income countries. Agriculture, Nutrition and Health Academy Food Environments Working Group (ANH-FEWG). Innovative Methods and Metrics for Agriculture and Nutrition Actions (IMMANA) programme. London, UK.

- We have changed our title based on recommendations of the format from the Editor. The new title is clearer on the study design and will avoid the confusion identified above: *Investigating foods and beverages sold and advertised in deprived urban neighbourhoods in Ghana and Kenya: A cross-sectional study.*

ABSTRACT:

- Line 8, p. 2: I suggest replacing the text with "food environment of three deprived neighbourhoods in African cities"

- We have changed our study aim to be closer to our title. It now reads: *The aim of this study was to characterise the local foods and beverages sold and advertised in three deprived urban African neighbourhoods (p2/abstract).* We have also updated the later occurrence in the introduction (p5). This allows for a more consistent message throughout.

INTRODUCTION

- Line 57-59, p. 4: I would recommend to review the following statement "or indeed in low- and middle-income countries in general". Although most part of the evidence is from high-income countries, studies are increasingly exploring urban food environments in middle-income settings. See a few examples provided in the comment below.

- We have rewritten the statement in line with your comments. It now reads: "*There are few studies from Africa (mainly concentrated in South Africa (18)), although studies are increasingly exploring urban food environments in middle-income countries which have some parallels.(19,20)*" (p5)

- Line 20-22, p. 5: I suggest referring to studies carried out in other lower-income countries which already measured the types of foods sold by outlets. See below a few examples from Brazil:

1) Leite et al. 2018. Association of neighbourhood food availability with the consumption of processed and ultra-processed food products by children in a city of Brazil: a multilevel analysis. *Public Health Nutr.* 2018 Jan;21(1):189-200.

2) Duran et al. 2013. Neighborhood socioeconomic characteristics and differences in the availability of healthy food stores and restaurants in Sao Paulo, Brazil. *Health Place.* 2013 Sep;23:39-47.

3) Leite et al. 2012. Availability of processed foods in the perimeter of public schools in urban areas. Availability of processed foods in the perimeter of public schools in urban areas. *J Pediatr (Rio J).* 2012;88(4):328-34.

- Thank you for directing us to these papers that we were not aware of. We have updated our paper throughout with reference to Leite et al 2018 and Duran et al. 2012 (these were both useful in a couple of sections to reinforce points; refs 19-20). We did not include the reference for Leite et al. 2012 as the paper is in Portuguese and none of the authors in our study could read it to confirm how the study fits our work.

- Lines 38-40, p. 5: I would recommend including the main hypotheses of the study/research question at the end of this section.

- We have no pre-existing hypotheses as our study was exploratory and focused on primary data collection. We have opted to keep the paper reporting our aim only, which is appropriate in common scientific investigation.

METHODOLOGY

- Line 17-20, p. 6: Please could you provide more details on the selection of neighbourhoods with local community groups? Which kind of groups? What were the particular characteristics of these groups in each of the selected cities?

- We have added greater information about the selection of neighbourhoods in Appendix A (signposted to the reader in the methodology on p6). We did not have any further information available about our populations of interest than what we have mentioned in the paper.

- Lines 22-23, p. 6: Please give more information on the pilots (e.g. how many outlets were visited, was the tool validated for each setting?)

- We have expanded on the details here, although please note that we did not record specific numbers: *“The data collection method was piloted in each setting through surveying smaller areas close to where data collection would take place. The tool was tested on each outlet types (number of occasions for each depending on prevalence in an area) and further amendments were made to the tool.”* (p6)

- Lines 29-30, p. 6: Why have you decided to use a classification discussed during a project workshop instead of other tools? How were the categories chosen and who was involved in this workshop?

- We have added some detail here in Appendix B now: *“Outlet and advertisement types were defined following a project meeting involving all of the international project partners, researchers and local field workers representing both Ghana and Kenya. The aim was to find consensus over our definitions based on individual expertise and evidence from the wider literature (summarised during discussions), as well as local subject knowledge. Definitions were then validated and refined during the pilot phase of our tool. We opted against using other existing classifications since they were often derived for other settings or countries that were not always relevant to the contexts we were collecting data in. Through designing our own classification, we developed a new system that was relevant to urban Ghana and Kenya, simple and efficient for data collection, and comparable for both countries.”*

- Line 36, p. 6: Which parameters/references were used to group foods expected to increase or decrease in consumption following the nutrition transition? Please clarify and give more detail in the methodology section.

- We have made this clearer in Appendix B: *“We also grouped foods (based on expert opinions and evidence across the literature) into whether we would expect them to increase or decrease in consumption following the nutrition transition to situate our data within broader nutritional trends in African countries.”* (p4-5).
- We have also added details about how we identified foods as healthy or not based on comments from the other reviewer in Appendix B (p5).
- We also added some more detail to set up the context of the nutrition transition at the start of the paper to help contextualise this to the reader: *“The generalised trends in dietary changes, termed the nutrition transition, include increased consumption of fat, particularly vegetable and edible oils; increased added sugar; increased animal-source foods and decreases in cereals and fibre, specifically in coarse grains, staple cereals and pulses.”* (5,6) *The exact*

nature of changes in dietary behaviours, and the foods that drive the nutrition transition, vary according to region.(7)" (p4)

- Lines 29-30, p. 7: Please include the name and reference of the Statistical Software used in the analysis.

- We have included this now: "*Analyses were completed using R statistical software.*" (p7)

- Line 20, p. 8: Include the total of all formal outlets

- We have added in the total within Table 1 (p8-9).

- Lines 55-57, p. 8: Although the authors say: "outlets on average offered a mixture of healthy and unhealthy foods", Table 1 does not show which outlet types were more responsible or not for healthy or unhealthy food availability. It would be interesting to see the frequency of food types per outlet types (e.g. a table presented in the appendixes).

- Agreed – we have added some detail about this into Appendix C now (specifically Table C5). We refer in the main document to this material as well: "*(Table 1; see Appendix C for more detail)*" (p9).
- We also added into Appendix B (p5) how we grouped foods as healthy or not to make it clear about our general interpretation of the results and help guide the reader.

- Lines 15-55, p. 9: Again, it would be interesting to see the types of foods advertised per outlet types (instead of per cities) as well (e.g. as an appendix).

- Agreed – we have added some detail about this into Appendix C now (specifically Tables C6 and C7). We refer in the main document to this material as well: "*(Table 2; see Appendix C for more detail)*" (p10).

DISCUSSION

- Lines 15-19, p. 13: I would suggest the authors to expand on the consequences that this result could pose to the health system. 1) poor dietary intake and alcohol abuse are among key NCD risk factors, 2) NCDs are the leading cause of death globally, killing more people each year than all other causes combined, 3) 80% of NCDs occur in low- and middle-income countries.

- Thank you. We have now added the following text to state the importance of our study in line with your suggestions: "*These findings require attention since the availability and promotion of unhealthy energy-dense nutrient poor foods within deprived urban African environments may place considerable burdens on health systems. Poor dietary intake and alcohol abuse are key drivers of NCDs accounting for 73.4% of all deaths globally.(4) With ~80% of NCDs occurring in low and middle income countries and an increasing burden in sub-Saharan Africa,(3) the consequences of unhealthy environments are potentially considerable.*" (p13)

- Lines 40-42, p. 13: The authors should be present a few examples and compare with the literature.

Although most of interventions to improve availability of healthier options in the community food environment are from high-income settings, there is still an opportunity to cite a few examples from the existing literature. E.g. Gittelsohn, et al. The Impact of a Multi-Level Multi-Component Childhood Obesity Prevention Intervention on Healthy Food Availability, Sales, and Purchasing in a Low-Income Urban Area. *Int J Environ Res Public Health*. 2017 Nov; 14(11): 1371.

- Thank you. We have included the reference you suggested to provide an example (see p13; ref 34).

- Lines 33-38, p. 14: The authors could cite a few examples of increased investments from the SSBs multinationals in the African region (e.g. Taylor et al. 2016. Carbonating the World: The Marketing and Health Impact of Sugar Drinks in Low- and Middle-income Countries. Available at: <https://cspinet.org/sites/default/files/attachment/Final%20Carbonating%20the%20World.pdf>)

- Thank you. We have included the reference in the text (see p14; ref 39).

- Line 44, p. 14: There are more recent references that could be used here (this one from 2011 is out of date). I recommend the authors to review a few citations and try to use more recent ones.

- We have added a more recent study here based on your suggestion (see p14; ref 40).

CONCLUSIONS

- Lines 46-51, p. 16: "The physical built environment is largely ignored aspect...". Do the authors refer to the African context? There is extensive literature and food policy interventions looking at this aspect in high- and middle-income settings (See the NOURISHING database, available at <https://www.wcrf.org/int/policy/nourishing-database>)

- We have added "*in the urban African context*" (p16) here so it is clear what we are referring to. We do note that our introduction makes reference to the large amount of literature in high-income countries.

VERSION 2 – REVIEW

REVIEWER	Larissa Mendes Universidade Federal de Minas Gerais
REVIEW RETURNED	26-Feb-2020

GENERAL COMMENTS	The suggestions and questions were answered.
--

REVIEWER	Fernanda H Marrocos Leite Center for Epidemiological Research in Nutrition and Health, School of Public Health, University of São Paulo
REVIEW RETURNED	21-Feb-2020

GENERAL COMMENTS	** Strengths and Limitations of this Study - Line 26, p. 3: Replace "food types sold" with "food and beverage" - Line 28, p. 3: Replace "foods sold" with "foods and beverages sold"
--

	** Introduction  - Line 24, p. 5: replace "types of foods sold" with "types of foods and beverages sold" - Lines 28-29, p. 5: The following sentence is confusing, please review it "These narrow conceptualisations of food environments limit our understanding of the food environment." - Line 31-34, p. 5: I suggest the authors to delete the sentence "approach to measuring the food environments" so it reads: "We extend previous approaches by utilizing a multi-dimensional approach to measuring the location and types of food outlets and adverts, as well as what foods and beverages are being sold or advertised." ** Statistical analysis  - Line 43, p. 7: Please include the version that was used and respective reference. ** Results:  - Line 22 p. 9: Please replace "healthy and unhealthy foods" with "healthy and unhealthy products (or items)" - so it also account for beverages. Also replace "Table 1" (line 22) with "(Tables C5 and C6)". - Line 36, p. 13: Replace "unhealthy foods" with "unhealthy foods and beverages" or "unhealthy products" - Lines 37-38: The authors should review this statement: "Formal outlets may be amenable to planning regulations and interventions to promote healthier foods" - shouldn't it be the other way around? Informal outlets may be? This paragraph is confusing - please review it. General comments:  ** Table titles: Please review titles from all Tables presented as an annex, to incorporate "beverages" as well (Ex: Table C3: Percentage of foods sold between formal and informal outlets - it should read "Percentage of foods and beverages sold") ** The authors should consider improving the standard of written English through copy-editing and proofreading.
--	--

VERSION 2 – AUTHOR RESPONSE

Reviewer(s)' Comments to Author:

Reviewer: 1

Reviewer Name: Larissa Mendes

Institution and Country: Universidade Federal de Minas Gerais

Please state any competing interests or state 'None declared': None

Please leave your comments for the authors below

The suggestions and questions were answered.

Reviewer: 2

Reviewer Name: Fernanda H Marrocos Leite

Institution and Country: Center for Epidemiological Research in Nutrition and Health, School of Public Health, University of São Paulo

Please state any competing interests or state 'None declared': None declared

Please leave your comments for the authors below

- Thank you for capturing the minor changes that we missed originally.

** Strengths and Limitations of this Study

- Line 26, p. 3: Replace "food types sold" with "food and beverage"

- Line 28, p. 3: Replace "foods sold" with "foods and beverages sold"

- These have been both changed as suggested (see p3)

** Introduction

- Line 24, p. 5: replace "types of foods sold" with "types of foods and beverages sold"

- We have made this change (see p5)

- Lines 28-29, p. 5: The following sentence is confusing, please review it "These narrow conceptualisations of food environments limit our understanding of the food environment."

- We have revised the sentence to : "Only focusing on outlet type alone and ignoring these broader characteristics limits our ability to build detailed measures of food environments to truly assess their influences on people" (p5)

- Line 31-34, p. 5: I suggest the authors to delete the sentence "approach to measuring the food environments" so it reads: "We extend previous approaches by utilizing a multi-dimensional approach to measuring the location and types of food outlets and adverts, as well as what foods and beverages are being sold or advertised."

- We have made the suggested change (p5)

** Statistical analysis

- Line 43, p. 7: Please include the version that was used and respective reference.

- We have made the changes suggested (p7)

** Results:

- Line 22 p. 9: Please replace "healthy and unhealthy foods" with "healthy and unhealthy products (or items)" - so it also account for beverages. Also replace "Table 1" (line 22) with "(Tables C5 and C6)".

- We have just replaced the foods bit with 'items' (p9) in line with your suggestion
- We opt to not make the revision to 'Tables C5 and C6' simply because it ignores the other relevant tables within Appendix C (e.g. differences between formal and informal outlet types overall). Since this information was requested in the first round of reviews, we do not wish to overlook it now in the second round. The change is only minor and does not detract from the message of the paper. We have removed Table 1 from the sentence as you suggest (p9).

- Line 36, p. 13: Replace "unhealthy foods" with "unhealthy foods and beverages" or "unhealthy products"

- We have made the suggested change (p13)

- Lines 37-38: The authors should review this statement: "Formal outlets may be amenable to planning regulations and interventions to promote healthier foods" - shouldn't it be the other way around? Informal outlets may be? This paragraph is confusing - please review it.

- Informal outlets are less amenable to planning regulations given that a local vendor who sells products on a mat on the floor can easily move. Formal outlets have an established structure that is less mobile. The view is common in the African context, but also follows our findings (formal outlets sell more healthier products compared to informal outlets so we should intervene with them).
- To avoid confusion, we have included examples of potential policies here now to support the justification for tackling unhealthy formal outlets and clarified the above point: "Given that they have physical structures (unlike informal outlets), formal outlets may be amenable to planning regulations (e.g. restricting location, rent subsidies) and interventions to promote healthier foods (e.g. price subsidies, promoting healthier products)" (p13)

General comments:

** Table titles: Please review titles from all Tables presented as an annex, to incorporate "beverages" as well (Ex: Table C3: Percentage of foods sold between formal and informal outlets - it should read "Percentage of foods and beverages sold")

- We have reviewed and added in 'beverages' through the appendices now.

** The authors should consider improving the standard of written English through copy-editing and proofreading.

We have made minor changes to the written English in both the main paper and supplementary appendices so improve the clarity of our text.

VERSION 3 – REVIEW

REVIEWER	Fernanda H Marrocos Leite School of Public Health, University of Sao Paulo (USP)
REVIEW RETURNED	31-Mar-2020
GENERAL COMMENTS	No further comments